# Does Relative Age Influence Organized Sport and Unorganized Physical Activity Participation in a Cohort of Adolescents?

**DOI:** 10.3390/sports10070097

**Published:** 2022-06-23

**Authors:** Kristy L. Smith, Mathieu Bélanger, Laura Chittle, Jess C. Dixon, Sean Horton, Patricia L. Weir

**Affiliations:** 1Department of Kinesiology, University of Windsor, Windsor, ON N9B 3P4, Canada; chittlel@uwindsor.ca (L.C.); jdixon@uwindsor.ca (J.C.D.); hortons@uwindsor.ca (S.H.); weir1@uwindsor.ca (P.L.W.); 2Department of Family Medicine, Université de Sherbrooke, Sherbrooke, QC J1H 5H3, Canada; mathieu.belanger@umoncton.ca; 3Centre de Formation Médicale du Nouveau-Brunswick, Moncton, NB E1A 3E9, Canada

**Keywords:** relative age effects, participation trends, longitudinal, sport context, physical activity

## Abstract

Despite their prevalence, the longitudinal impacts of relative age effects (RAEs) on sport and other forms of physical activity (PA) are understudied. This study examined longitudinal participation patterns in organized sport (team and individual), unorganized PA, and non-participation with respect to RAEs in a prospective cohort of adolescents. Data from the first 24 cycles of the MATCH study were used for analyses. Elementary students (*n* = 929) were recruited from 17 schools in Atlantic Canada. Respondents self-reported PA three times/year. Mixed multilevel logistic models compared the likelihood of participating in each context across birth quarter. Chronological age and gender were considered, along with the interaction between chronological and relative age. Individuals born in Quarter 1/Quarter 2 were more likely to report participation in organized team sport but not individual sports. Relatively older participants born in Quarter 2 were more likely to report participation in unorganized PA. Increasing chronological age was associated with decreased participation in organized sport (particularly team-based) and increased non-participation. Gender was not associated with organized sport participation, but girls were under-represented in unorganized PA and more likely to report non-participation. The interaction parameters suggested that RAEs were consistent throughout adolescence in each context. Longitudinal analyses suggest RAEs are context dependent.

## 1. Introduction

Despite efforts to communicate the importance of regular physical activity, high levels of physical inactivity remain a global concern [1]. Adolescence is a particularly concerning period as it is often marked by significant declines in physical activity participation [2,3,4]. “Physical activity” encompasses a variety of activities such as competitive sport, leisure walking, and occupational and domestic activities, among others [5,6]. Consequently, context is an important consideration when evaluating physical activity patterns [7]. Likewise, recommendations for increasing physical activity can incorporate different domains such as active transport (e.g., walking to school), participation in unorganized sport or active play [8], and even breaking up sedentary time with light intensity activity [9]. While participation in organized sport has been associated with a diverse range of positive outcomes among youth (e.g., enhanced resilience, character development, social skills [10,11]), the health benefits of physical activity (e.g., cardiorespiratory fitness, improved posture) might also be obtained in an unorganized context [12,13]. Thus, both organized and unorganized activity provide beneficial opportunities to increase participation. While the work to date by Bengoechea et al. [14] and Hardy et al. [15] present equivocal findings with respect to whether participation is higher in organized or unorganized activities, it is clear that through adolescence, both categories of participation are important contributors to overall activity level [16]. 

One determinant that has been associated with participation in the physical activity context of organized sport is relative age. Relative age refers to the difference in age between individuals in a cohort [17,18], such as those grouped within the same grade in the school system. These age groupings are intended to help promote developmentally appropriate instruction and competition [19,20]. However, a one- or two-year age difference can result in significant physical, psychological, and experiential differences, particularly at young ages when a few months can represent a considerable proportion of a child’s lived experience. In addition to experiential differences, age differences within a given age grouping can include considerable variability in biological maturity from childhood through to adolescence [18,21]. 

The term relative age effects (RAEs) refers to the (dis)advantages commonly associated with the aforementioned age groupings [22], providing a benefit to those who are relatively older (meaning that they were born closer to but following an arbitrary cut-off date) and disadvantaging those who are relatively younger in the same age group or cohort. With respect to sport, the advantage conveyed to those who are relatively older is assumed to be present when an over-representation of relatively older players is observed among sport participants (e.g., on a specific team), particularly at elite levels. While RAEs have predominantly been studied in team sports such as ice hockey, soccer, volleyball, and basketball (see Cobley et al. [23] for a review), individual sports, where success is predicted by strength or endurance capabilities, such as tennis [24], cross-country and alpine skiing [25], and sprinting [26,27], may also be affected. Potential consequences of RAEs include relatively younger players dropping out of organized sport at an earlier age [28,29,30] or adapting by taking up forms of activity that are less structured [31] or less competitive [32] in nature. To date, RAEs have primarily been found in male sport contexts ([23]), but a meta-analysis of RAEs in female sport shows a pervasive effect there as well (see Smith et al. [33]). Sex is nevertheless believed to be a significant moderator of the effect (Baker et al., 2010), with previous findings suggesting RAEs in females are more variable and weaker in magnitude when compared to males [23,34]. 

Several explanations and theoretical models of RAEs have been presented over the years. For instance, many relative age studies published to date have focused on maturation and selection (e.g., [20,23,35]). Being relatively older is often associated with greater physical and anthropometric maturity (e.g., height, weight, strength) [36]. During the maturational period, up to and including puberty, relatively older individuals may be viewed as more talented on the basis of their more advanced physical and anthropometric development. As a result, relatively older sport participants may benefit from a higher likelihood of being selected to elite teams, providing enhanced access to coaching and facilities, thereby facilitating greater skill development than their relatively younger peers. 

Hancock et al. [37] argue that RAEs are largely tied to social influences. Specifically, RAEs are attributed to early experiences of success and the manner by which expectations of coaches and parents influence athletes’ behaviors and self-perceptions, which ultimately impacts their level of achievement. Finally, a model proposed by Wattie et al. [38] suggests that the impact of relative age comes at the intersection of individual (e.g., gender, relative age), task (e.g., team or individual sport, organized vs. unorganized activity), and environmental constraints (e.g., age grouping policies) within developmental systems theory. Relative age can be viewed as one of many factors influencing the probability of sport participation, and there is opportunity for change throughout development. This developmental perspective is adopted within the current longitudinal study. 

Relative age research has largely been cross-sectional in nature and focused on performance, examining the athlete population of a specific sport (e.g., basketball [39]) or sporting event (e.g., Olympic Games [40]). While this strategy has provided useful descriptions of the prevalence of RAEs, it does not inform about the potential longer-term effects of relative age advantages and/or disadvantages on sport and/or physical activity participation for children and adolescents. Furthermore, evidence pertaining to RAEs in sport predominantly emerges from studies in organized sport and competitive contexts and is less commonly studied in recreational sport [23]. Observations from the few studies published to date suggest that RAEs may be highly variable within recreational contexts and influenced by a variety of factors and/or constraints [41,42,43]. Furthermore, published studies concerning the role of RAEs on outcomes associated with participation in unorganized sport and/or physical activity in children and adolescents are virtually non-existent. 

Relative age effects have also been observed in a variety of areas within the school system such as physical education [44], high school leadership activities [45], standardized test scores (math and science [46]), university attendance [46], and emotional regulation [47]. These findings suggest that comparisons with peers occur in other contexts and could potentially impact the overall course of development for youth [48]. If relatively younger youth are habitually disadvantaged in organized activities and have fewer opportunities to benefit from sport system development, they could suffer from reduced perceptions of competence and eventually disengage from a physically active lifestyle. For example, perceived competence in a physical education setting has been positively associated with leisure-time physical activity (e.g., [49,50]). The impact of relative age may be evidenced in children’s degrees of involvement not only in organized activities but also in unorganized activities or in a lack of participation altogether; these possibilities deserve further consideration. 

To address the aforementioned gaps in the literature, the primary purpose of the current study is to examine the longitudinal patterns of participation in organized sport, unorganized physical activity, and non-participation with respect to relative age, in a prospective study of adolescents. The impact of gender and chronological age are considered, along with the interaction between chronological age and relative age within each participation/non-participation context. We hypothesize that a greater proportion of those who are relatively older participate in organized sport (both team and individual), while those who are relatively younger gravitate toward unorganized physical activities (where talent selection activities that favor the relatively oldest are notably absent), or a lack of participation altogether. Furthermore, it is expected that the hypothesized trends will become more pronounced with increasing chronological age for boys as they enter adolescence and become less pronounced for girls in alignment with previous meta-analytical findings [23,33].

## 2. Materials and Methods

### 2.1. Respondents and Data Collection

To accomplish the research objectives, the database for the Monitoring Activities of Teenagers to Comprehend their Habits (MATCH) study was queried to examine the same respondents in multiple activities rather than a single sport in isolation, which is a noted limitation of previous work. The MATCH study is an ongoing longitudinal prospective study in the province of New Brunswick, Canada. The complete study protocol is provided in Bélanger et al. [51]. Briefly, a convenience sample of 19 schools were recruited, and it included both French- and English-speaking students from a range of socioeconomic neighborhoods located in both rural and urban areas. Two of the 19 schools were excluded due to a low return rate of consent forms (below 50%). All grade five and six students from the remaining 17 schools were invited to participate in the study. Respondents ranged from 10 to 12 years of age at the time of recruitment, representing a critical timeframe when physical activity levels are often found to attain a peak before typically experiencing a decline as the transition from childhood to adolescence occurs [2,52]. Initial recruitment in the fall of 2011 included 802 of 1545 eligible students (response proportion of 52%), and other students from the participating schools were allowed to join the study in subsequent cycles (total number of respondents = 929). Please note within this study, the term ‘respondent’ refers to any individual who participated in the MATCH study, ‘survey response’ indicates participation in a single MATCH data collection cycle, and ‘participant’ indicates reported participation in sport and/or physical activity (outlined further below). 

The same respondents were asked to complete self-report questionnaires three times per year (fall, winter, and spring) at four-month intervals, which were combined to capture seasonal variations in one year of physical activity participation [6,51,53]. The first eight years of the MATCH study were used for the current analysis (*n* = 24 cycles). The questionnaires were completed under the supervision of a trained research assistant in a classroom setting, which was scheduled at a time that was convenient for the teacher. The initial session was 45–60 min in duration, and subsequent questionnaires were completed in 20–30 min. Respondents answered questionnaires in the first language of instruction at the school. 

### 2.2. Measures

Respondents reported all forms of leisure time physical activity over the preceding four months from a list of 36 activities commonly practiced by youth in Atlantic Canada [54], which was similar to other validated physical activity checklists [55,56,57]. Respondents could also report additional activities under “other”. The students were specifically instructed to disregard physical activity taking place during physical education classes because youth do not have control over the content of these classes [58]. The frequency of participation in each activity was collected using the response options “never”, “once per month or less”, “two to three times per month”, “once per week”, “two to three times per week”, “four to five times per week”, and “almost every day”. Respondents were also asked to report “with whom” they had most often engaged in each activity by selecting from “by myself”, “organized group or team”, “sibling(s)”, “friend(s)”, or “parent(s)”. The answer to this question was considered to be an indicator of the context in which each activity was most often practiced [59]. Respondents were described as ‘participants’ in the physical activity context being examined if they reported participating in any activity pertaining to each category “once per week” or more, similar to previous studies [14,60,61], thereby creating a binary measure (participant/non-participant). 

For each of the cycles in the longitudinal sample, the classification of “relatively older” versus “relatively younger” must be made with reference to the cut-off date used to group individuals within the specific sporting activities. The dates used for activities recognized as “sports”, defined by the existence of a recognized, provincial organization registered with Sport New Brunswick (SportNB), were obtained by contacting each respective organization listed in SportNB’s online directory. If no response was received from the provincial organization, the national organization was contacted through the Sport Canada website. The majority of sports employed a 31 December cut-off date to group athletes. Other organized activities use cut-off dates as well, including the education system, which utilizes 31 December in the province of New Brunswick [62]. Therefore, the decision was made to code all respondents using this date because it is the most commonly encountered age grouping experienced by the respondents in this study. The coding of birth quarter proceeded as follows: Quarter 1—January to March; Quarter 2—April to June; Quarter 3—July to September; Quarter 4—October to December. Thus, the relatively oldest were identified as ‘Quarter 1’ subsequently followed by Quarter 2, Quarter 3, and Quarter 4 (i.e., the relatively youngest). This method of coding is consistent with previous studies (e.g., [63,64,65]).

Similar to other studies employing the MATCH sample, respondents were classified as “organized sport participants” if they reported participation with an “organized group or team” [61] in one of the recognized sports in New Brunswick. Organized sports were further categorized as “organized team sports” if the sport is most often practiced or played with more than one participant competing at the same time or multiple participants are a necessary element for competition. Alternatively, organized sports were classified as “organized individual sports” if participation most often occurs with only a single competitor competing at any given time. These definitions were consistent with previous research [58,61]. Organized team sports that met the criteria outlined above included: ice hockey, ringette, baseball/softball, basketball, soccer, volleyball, and handball. Organized individual sports that met these criteria included: bicycling, track and field, jogging or running, badminton, tennis, downhill skiing or snowboarding, and cross-country skiing. The specific activity identified by the respondent was not important but rather the classification of “participant” in a particular context (i.e., organized team and/or individual sport).

To ensure accurate representation of each activity, sports that used a cut-off date other than 31 December were excluded to prevent bias from the use of an alternative date (i.e., 1 July for ice skating and 1 August for golf). Gymnastics was omitted because a reversal of the classic relative age pattern would be expected in this context. Late maturation is valued in aesthetic sports with peak performance occurring prior to the completion of maturation [38,66]. Accordingly, an over-representation of relatively younger participants and an under-representation of relatively older participants might be expected (i.e., reversal of the typical RAE pattern). This expectation was supported in the current data and therefore, gymnastics was removed from the organized sport analysis for this study to prevent an underestimation of the typical RAE present in this sample. Swimming was excluded because participants are categorized based on their chronological age on the day of competition. Other activities were excluded because the cut-off was unknown (i.e., football, boxing/wrestling), or multiple activities were grouped together (i.e., canoe/kayak and karate/judo/tai chi/taekwondo) in the questionnaire and could not be categorized consistently as an organized sport due to the absence of a provincial governing body for one or more of the activities. 

Using the classification system derived by Ward et al. [61] and validated by MacKenzie and colleagues [67], five of the 36 activities were categorized as “unorganized physical activity”, regardless of the context in which they were most often practiced, including trampoline, jump rope/skipping, games (e.g., chase, tag, hide and seek), home exercise, and weight training. Indoor chores and outdoor chores were excluded given the non-volitional and low intensity nature of these activities. Participation in the remaining activities was also considered to be unorganized if the individual reported to most often practice it alone, with friends, with siblings, or with a parent. A respondent who did not participate in any activity within a classification category was considered a “non-participant”. A summary of the classifications is available in Appendix A. 

### 2.3. Data Analysis

Prior to analyses, missing data for birth date (0.002%) was reviewed and corrected across all 24 cycles. A total of 12,061 survey responses were available for examination. Respondents (*n* = 929) participated in an average of 13 survey cycles each. Three objectives guided statistical analyses. First, the relative age trends in each of the five participation/non-participation contexts were assessed (i.e., any organized sport, organized team sport, organized individual sport, unorganized physical activity, and non-participation). Second, the impact of gender and chronological age were considered for each of the five contexts. Gender was self-reported by respondents in the MATCH study. In seven survey responses, (<0.01%) the respondent selected “other” as their gender category. These instances could not be included in the analyses due to insufficient sample size. Third, the longitudinal interaction between chronological age and relative age was considered within each participation/non-participation condition. To accomplish these objectives, a mixed multilevel logistic model was conducted for each of the five participation/non-participation contexts. The potential for intra-class correlation because of repeated measures among respondents and school-level clustering was accounted for by including random intercepts for these variables. Birth quarter (representing relative age), chronological age (by year – included as a continuous variable), and gender were included as independent variables, along with the interaction between chronological age and relative age. The fourth quarter (Quarter 4) was used as the comparison group. The interaction did not meet the criteria for statistical significance (*p* > 0.05) in any of the participation/non-participation contexts. Thus, the models presented in this paper do not include estimates for interaction parameters in the interest of parsimony. 

## 3. Results

Our primary findings with respect to relative age trends, as indicated by the representation of participants in each birth quarter, are outlined below. Results are presented for organized sport (overall, team, and individual), unorganized physical activity, and non-participation, and they include findings for gender, chronological age, and the interaction between chronological age and relative age in each respective context. Of the participation contexts examined, unorganized PA had the most participants, and organized individual sport had the lowest number of participants throughout adolescence (Table 1). In general, there were more boys than girls reporting participation in all types of sport/PA. 

### 3.1. Organized Sport

In comparison to those born in Quarter 4, those born in Quarter 3 were less likely to report participation in any organized sport overall, *t* (976) = −2.205, *p* < 0.05 (Table 2a). However, when team and individual sport were analyzed separately, Quarter 1- and Quarter 2-born participants (i.e., the relatively oldest) were more likely to report participation in organized team sports when compared to Quarter 4 (i.e., the relatively youngest), *t* (1093) = 1.995, *p* < 0.05 and *t* (1003) = 2.138, *p* < 0.05, respectively (Table 2b). No statistical differences based on relative age were noted for organized individual sport participation suggesting equal representation of participants from each birth quarter (Table 2c). 

Gender was not observed to be an important variable for any of the organized sport participation contexts (overall, team, or individual). Negative estimates for chronological age indicated that as participants became older, their participation in organized sports overall, *t* (915) = 1.356, *p* < 0.05, and organized team sports, *t* (3643) = −2.801, *p* < 0.01, declined but their reported participation in organized individual sport was maintained. The lack of statistical significance for any of the interaction parameters suggest that RAEs observed for organized sport participation overall, as well as for team and individual sport were consistent throughout adolescence (i.e., the same patterns of over-representation by birth quarter were observed across the years examined).

### 3.2. Unorganized Physical Activity

Participants born in the second quarter (i.e., relatively older) were found to be statistically over-represented compared to the fourth quarter (i.e., relatively younger) in unorganized physical activity participation, *t* (1544) = 2.731, *p* < 0.01 (Table 3). 

Participation in unorganized physical activity did not change with chronological age, but we noted that girls were less likely than boys to report participation in an unorganized context, *t* (1473) = −2.850, *p* < 0.01. The test of the interaction suggested that relative age trends (by year) were invariant in unorganized physical activity participation across the years examined. 

### 3.3. Non-Participation

With respect to non-participation (i.e., individuals who did not report participation in an organized and/or unorganized context), Quarter 2-born individuals were found to be significantly under-represented compared to those born in Quarter 4, *t* (1556) = −3.374, *p* < 0.01 (Table 4). 

Non-participation increased with chronological age, *t* (5216) = 2.657, *p* < 0.01, and there was an increased likelihood of non-participation among girls versus boys, *t* (1482) = 2.012, *p* < 0.05. The longitudinal trends (by year) with respect to relative age were stable in the non-participation category, as indicated by the lack of statistical significance in the test of the interaction. 

## 4. Discussion

The longitudinal analyses undertaken in this study were based on eight years of data spanning late childhood to late adolescence. The primary purpose was to examine participation with respect to relative age in five participation/non-participation contexts. Secondary objectives included consideration of gender and chronological age as well as the interaction between chronological age and relative age. We found that relatively older (Quarter 1 and Quarter 2) respondents were more likely to report participation in organized team sport when compared to the relatively youngest (Quarter 4). This finding is consistent with past research and supports our hypotheses for the organized team sport context. However, in contrast to previous findings (e.g., [24,25,26,27,33]) and our hypotheses, no relative age trends were observed for organized individual sport participation. Furthermore, RAEs appeared to be associated with participation in unorganized physical activities, with Quarter 2-born participants over-represented. Notably, gender did not play a role in organized sport participation, but girls were under-represented in unorganized physical activities and more likely to report non-participation. Increasing chronological age was associated with decreased participation in organized sports overall and specifically in team-based contexts (but not individual sports); it was also associated with increased non-participation in this sample. Relative age effects appeared to be consistent throughout the developmental years examined in this study (i.e., the interaction did not meet the criteria for statistical significance (*p* > 0.05) in any of the participation/non-participation contexts). 

An over-representation of relatively older participants in team-based sports supports the role of talent selection processes in perpetuating RAEs [23]. “Tryouts”, whereby coaches and scouts scrutinize athletes’ skills for the purpose of identifying the most talented individuals, are often required to gain membership on an organized team. These activities are believed to favor those who are relatively older, as they are likely to be taller/stronger and have acquired more life experience as a result of being chronologically older [18]. Once membership is gained, these selected individuals continue to benefit from being relatively older by having access to higher levels of training, coaching, and competition [20], resulting in an accumulated advantage from initial (albeit subtle) differences in growth, development, and experience [23,68]. 

Evidence for RAEs in individual sports has been less consistent in comparison to team sports [25]. Previous reports of RAEs have emerged from studies of individual sports for which physical attributes might provide an advantage such as tennis [24], cross-country and alpine skiing [25], and sprinting [26,27]. However, the examination of sport contexts that rely more predominantly on technical skills (vs. physical prowess) have not produced equivalent findings (e.g., golf [69], shooting sports [70]). It has been theorized that RAEs may be less prominent in individual sports due to comparisons between peers occurring after performance, and they are often informed by more objective scores such as race time or a deductive rating system administered by a group of judges. In contrast, the peer-to-peer comparisons associated with team selection processes are often made in the midst of competition or scrimmages where physical differences might be more apparent on the field of competition or during practice [25]. While “individual sport” in this study included several activities associated with physical demands, the lack of evidence for biased birth distributions suggests this type of sport context may be more likely to provide equitable opportunities for success. However, the compiled sample included a variety of sports, and each respective sport would need to be examined in detail to identify exact trends along with sport-specific factors contributing to the existence of RAEs or lack thereof. It is also important to note that several individual sport contexts were not available for examination, and competition/skill level was not considered in these analyses (i.e., a stronger risk of RAEs would be expected at elite levels of performance and may be somewhat diminished in a heterogenous sample) [23,33]. 

Gender (or sex, as reported in some relative age studies) has been identified as a significant moderator of RAEs in previous research, with male RAEs reported to be greater in magnitude versus the female effect [23], even when participation numbers are similar [34]. However, reported gender did not appear to modify trends observed in this longitudinal sample. This is not completely unexpected or incongruent with previous findings, given that RAEs were primarily associated with team-based sport in these analyses and likewise have been consistently identified in cross-sectional samples of both boys and girls in other team-based contexts (e.g., [23,33]). Furthermore, while the pattern of risk varies when samples are compared, RAEs are almost always present and expected to be present during the adolescent transition years, when variability in maturational differences is greatest [18,21]. 

An over-representation of Quarter 2-born participants (versus Quarter 4) was observed for unorganized physical activity participation. This finding may suggest that the impact of RAEs extends beyond the commonly examined context of organized sport, as talent selection processes would not be required to gain membership in these activities and engagement would be based on individual and/or parental volition. While direct comparisons for children and youth are not available in previous research for unorganized physical activity participation (Larouche et al. [71] reported findings for physical activity participation in a sample that included both adolescents and adults), this trend highlights the need to address developmental biases in both organized and unorganized contexts for this demographic given the importance of regular participation on health outcomes [12,72]. Reduced levels of cardiorespiratory fitness have been reported for the relatively youngest among children ages 9–12 years [73], and these findings may be associated with the extent to which these individuals feel confident about their ability to engage in various physical activity contexts.

An increased focus on the development of fundamental movement skills may support participation in unorganized forms of physical activity among the relatively youngest and could also support girls who were less likely to report participation in this context compared to boys. A future consideration is to examine whether basic movement skills (e.g., sprint, jump, throw, catch, balance) differ within a same-age cohort along with consideration of gender. Coincidentally, New Brunswick is one of the few Canadian provinces that require elementary schools to have a dedicated physical education specialist on staff [8], which could increase the likelihood that the development of basic movement skills is being promoted. These skills could potentially promote successful participation in a variety of physical activities across the lifespan and minimize impact of RAEs in an unorganized context [74,75]. It should also be noted that “unorganized” included a large number of activities (total of 36) in a variety of settings. Future studies should consider breaking this category down in order to identify any existing patterns in certain types of unorganized activity (e.g., group versus individual activities, domestic chores versus active transportation versus leisure pursuits).

“Non-participation” was defined as a lack of self-reported participation in an organized sport or unorganized physical activity at a minimum frequency of once per week, which is consistent with previous studies [14,60,61]. The relatively oldest participants born in Quarter 2 were less likely to report non-participation when compared to the relatively youngest born in Quarter 4. This further supports the notion that RAEs in organized contexts may potentially impact other aspects of health behavior during adolescence. Future work could examine whether this finding extends to sedentary behavior (e.g., screen time). An increased risk of non-participation was also observed for girls and with increasing chronological age. These findings are consistent with previous observations that activity levels begin to decline during adolescence (e.g., [76,77]) and should be taken into consideration when designing interventions to address non-participant behavior. 

Returning to the theoretical framework of Wattie et al. [38], we have demonstrated that different constraints may be operating that impact adolescent participation in activities across a broad range of contexts. Individual constraints of chronological age and relative age continue to be present within the realm of organized sport overall, and specifically team-based activities, while relative age and gender impact less structured activities. Task-related constraints (e.g., team versus individual, organized versus unorganized) also differentiate participation. From a practical standpoint, the Wattie et al. framework and the findings of this longitudinal analysis (and future work) should be used to develop strategies to support the relatively youngest in various contexts. It may be necessary to address constraints at all levels (individual, task, and environmental) beyond the realm of organized sport. For instance, schools and/or community organizations may want to promote extracurricular opportunities by grouping students according to birth quarter to minimize the impact of age-related disparities. Children and youth could also be grouped according to their level of competency in a given activity. Furthermore, interventions are required to overcome RAEs throughout adolescence, since the relative age trends observed in this study were consistent across the eight years examined, suggesting that the relatively youngest did not overcome participation disadvantages at any point during the adolescent transition years. 

The strengths of this study include that it is one of the first to provide a longitudinal examination of RAEs and it does so during the “pivotal” years of youth sport development [78]. Furthermore, the same respondents were evaluated in multiple contexts rather than in a single sport (e.g., team and individual organized sport, unorganized physical activity). The limitations of this study include those inherent in self-report questionnaires such as recall bias and social desirability bias, both of which could have resulted in misclassification and the over- or under-estimation of organized sport or unorganized physical activity participation. No objective measures of participation were utilized, and some organized sport contexts were not available for examination, or the cut-off date was unknown; however, the use of the questionnaire allowed for the investigation of several types of participation in addition to context. Furthermore, the study sample was not designed to be representative of a population, meaning that more research is needed to determine if the results of this study are generalizable to different youth populations [79].

## 5. Conclusions

Longitudinal analyses suggest RAEs are context dependent and persist through adolescence. The relatively oldest respondents (Quarter 1 and Quarter 2) were over-represented with respect to reported organized team sport participation, while individual sport contexts appeared to be unaffected, as evidenced by the lack of statistically significant differences in the birth distribution. Relative age inequities also appeared to be associated with participation in unorganized physical activities, with Quarter 2-born participants over-represented. Chronological age impacted organized sport participation overall and team-based contexts, while gender was observed to influence participation in unorganized physical activity and non-participation. Detailed research into specific constraints contributing to the observed patterns would be valuable. 

## Figures and Tables

**Table 1 sports-10-00097-t001:** Average proportion of respondents in each context across the 24 survey cycles included in the analyses.

Context ^1^	Girls (*n* = 511, 55%)	Boy (*n* = 418, 45%)	Total (*n* = 929)
Organized sport ^2^	37.2%	45.8%	41.9%
Team sport	31.8%	40.1%	35.6%
Individual sport	11.3%	13.6%	12.3%
Unorganized PA	79.0%	82.6%	80.6%
Non-participant	14.3%	12.0%	13.3%

^1^ Respondents could be included in organized sport and unorganized physical activity (PA) simultaneously, but the non-participant category is exclusive. ^2^ Participants included in organized sport overall may have reported team and/or individual sport participation.

**Table 2 sports-10-00097-t002:** Odds ratios of participation in organized sport across relative age, (**a**) overall; (**b**) team; (**c**) individual.

Fixed Effects	Odds Ratios ^1^	95% Confidence Intervals	
		Lower	Upper
**(a) Organized Sport (Overall)**			
Relative age (Q1 vs. Q4) ^2^	1.03	0.98	1.08
Relative age (Q2 vs. Q4)	1.04	0.99	1.09
Relative age (Q3 vs. Q4)	**0.95**	0.90	0.99
Gender (Girl vs. Boy)	1.02	0.99	1.06
Chronological age	**0.98**	0.96	1.00
**(b) Organized Sport (Team)**			
Relative age (Q1 vs. Q4)	**1.06**	1.00	1.13
Relative age (Q2 vs. Q4)	**1.06**	1.01	1.12
Relative age (Q3 vs. Q4)	0.95	0.90	1.01
Gender (Girl vs. Boy)	1.00	0.97	1.05
Chronological age	**0.97 ***	0.94	0.99
**(c) Organized Sport (Individual)**			
Relative age (Q1 vs. Q4)	1.02	0.98	1.07
Relative age (Q2 vs. Q4)	1.03	0.99	1.07
Relative age (Q3 vs. Q4)	0.97	0.93	1.01
Gender (Girl vs. Boy)	1.01	0.99	1.04
Chronological age	1.01	0.99	1.03

^1^ Statistically significant values are indicated with bold text (*p* < 0.05) and * (*p* < 0.01). ^2^ Birth quarter is identified by the following: Quarter 1 (Q1), Quarter 2 (Q2), Quarter 3 (Q3), and Quarter 4 (Q4).

**Table 3 sports-10-00097-t003:** Odds ratios of participation in unorganized physical activity across relative age.

Fixed Effects	Odds Ratios ^1^	95% Confidence Intervals	
		Lower	Upper
Relative age (Q1 vs. Q4) ^2^	1.02	0.98	1.05
Relative age (Q2 vs. Q4)	**1.04 ***	1.01	1.08
Relative age (Q3 vs. Q4)	1.03	1.00	1.06
Gender (Girl vs. Boy)	**0.97 ***	0.95	0.99
Chronological age	0.99	0.97	1.00

^1^ Statistically significant values are indicated with bold text (*p* < 0.05) and * (*p* < 0.01). ^2^ Birth quarter is identified by the following: Quarter 1 (Q1), Quarter 2 (Q2), Quarter 3 (Q3), and Quarter 4 (Q4).

**Table 4 sports-10-00097-t004:** Odds ratios of non-participation across relative age.

Fixed Effects	Odds Ratios ^1^	95% Confidence Intervals	
		Lower	Upper
Relative age (Q1 vs. Q4) ^2^	0.98	0.95	1.01
Relative age (Q2 vs. Q4)	**0.95 ***	0.93	0.98
Relative age (Q3 vs. Q4)	0.98	0.95	1.01
Gender (Girl vs. Boy)	**1.02**	1.00	1.04
Chronological age	**1.02 ***	1.00	1.03

^1^ Statistically significant values are indicated with bold text (*p* < 0.05) and * (*p* < 0.01). ^2^ Birth quarter is identified by the following: Quarter 1 (Q1), Quarter 2 (Q2), Quarter 3 (Q3), and Quarter 4 (Q4).

## Data Availability

The original data analyzed for this study are available through a data-sharing agreement with the MATCH study research team. More information on this may be obtained from the principal investigator of the MATCH study, Mathieu Bélanger.

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
