# Peer review of "Does Relative Age Influence Organized Sport and Unorganized Physical Activity Participation in a Cohort of Adolescents?"

_sports, 2022, doi:10.3390/sports10070097_

Round 1

Reviewer 1 Report

Personally, I would like to thank you the opportunity to review your manuscript. In general, this is an interesting research about RAE. I have some concerns throughout the manuscript. For that reason, I hope my recommendations will help you to improve the manuscript. First of all, just a curiosity, I would like to ask about the reason to use RA instead of RAE, you never include the term “effect” in your paper, but you discuss about the difference between BQ1 to 4.

Abstract:

Lines 10-11:

You wrote: “Despite their prevalence, the longitudinal impact of relative age effects (RAEs) on sport and other forms of physical activity (PA) are unknown”. There are a few papers relating RAE throughout time (Medic et al., 2007; Medic et al., 2009; Saavedra-García et al., 2019; Saavedra-García et al., 2016; Till et al., 2014; Wattie et al., 2012….)

Line 15:

“N.B., Canada” first time using an acronym, you need to explain, I suppose it means “Nouveau-Brunswick”

Line 16:

You use “quartiles”. Please, consider changing quartile by trimester, it is more easy and understandable, besides, there are three quartiles, and four trimesters, you use BQ1, BQ2, BQ3 and BQ4.

Lines 16-17:

You wrote: “Chronological age and gender were considered, along with the interaction between chronological and relative age”. This phrase is not easy to understand in abstract, but it makes sense only after reading the paper. Please, reconsider to explain it better.

Line 18:

You use BQ as birth quartile, but I suggest Q1, Q2, Q3 and Q4, only one paper (Skorski et al., 2016) use BQ after being revised over 600 papers about RAE. It is just a suggestion.

Line 22:

“but girls were under-represented” please include signification if possible.

Introduction

Line 32:

Citation 4. The survey is about 2012-13 season, and this paper was wrote 10 year later!, is this a problem?

Line 37:

Please consider to change “walking for exercise”

Line 59:

Cut-off date use to be January 1st.

Line 60:

It is a 10% in between children born January 1st (almost ten years) an children born in December, 32st. As you explain it is a 9,09%

Line 70:

“ice hockey and soccer”, really not enough, only 2 team sports. More sports are needed.

Line 93:

“Hancock et al.’s framework [38] argues” must start in a new paragraph.

Line 111:

Cite 24 is a very wide meta-analysis, it can be explained deeply.

Lines 135-137:

“To accomplish these objectives, the database for the Monitoring Activities of Teenagers to Comprehend their Habits (MATCH) study was queried to examine the same respondents in multiple activities rather than a single sport in isolation, a limitation of previous work.” I am sure this phrase is a part of material and methods, not introduction.

Materials and methods:

There are a lot of citations in this chapter, but not citations about methodological questions, I suggest to move this citations to the introduction chapter.

In general, you conduct multiple comparisons with the same data in function of different criteria. This multiple comparison can affect the p-values obtained in the results of your research. Anyway, it is acceptable.

Line 158:

“response proportion of 52%” this is a very serious problem, the sample can be easily biased. It must be clearly explained and included in the limitations of the study. Under my opinion, this is the critical point of the paper.

Lines 197-200

I cannot agree with the use of uniform distribution:

The assumption that the season of birth of the wider populations of different regions is uniform may be a valid assumption as sexual and reproductive life in humans is not generally subject to seasonal variation, although there may be minor effects due to urbanization, industrialization and religious or cultural traditions (Cowgill, 1966). However, it must be recognized that the assumption of uniform wider populations for all gender, region and age sub-groups is a limitation of the study. Assuming a uniform season of birth distribution in the wider population also overcomes the issue of variations in population season of birth distributions between the different calendar years in which the players were born. The different number of days in each month should also be taken into account (Doblhammer & Vaupel, 2001). Thus, the expected fraction of any group of players who were born in Q1 would be 90¼/ 365¼, compared with 91/365¼ in Q2 and 92/365¼ in each of Q3 and Q4 (same in Q5 to Q8) (Delorme y Champely, 2015; Doblhammer y Vaupel, 2001; Edgar y O'Donoghue, 2005, Saavedra et al. 2016).

I strongly suggest you take into account this correction to the uniform distribution, maybe you can obtain even better results in your research.

However, the worst solution is to assume the uniform distribution. As all the participants are from Canada, you can obtain expected values of QB in the national statistics from Canada.

Lines 225-226:

“in the questionnaire and could not be categorized consistently as an organized sport due to the absence of a provincial governing body for one or more of the activities” This is a serious problem to. I recommend as in line 158.

Lines 245-248:

“a mixed multilevel logistic model was conducted for each of the five participation/non-participation contexts. The potential for intra-class correlation because of repeated measures among respondents and school-level clustering was accounted for by including random intercepts for these variables” I really can’t find this results in your paper, nothing about multilevel logistic model. Can you detail this question?

Line 251:

BQ1 are usually used as reference BQ for comparison.

Results:

Line 256:

12,061 is the first time you inform about this number. This is, indeed, material and methods.

Table 1:

“Respondents could be included in organized sport and unorganized physical activity (PA) simultaneously” This is an important problem, the sum never gets 100%, always overpass this percentage, the intersection conjunct is not good for the paper and for the research. It would be really interesting to separate organized and unorganized PA, maybe any person practising organizes PA mustn’t be included in unorganized PA.

Table 2:

You include an “*” in chronological age “0.97*” but you did not explain what it means. It is important to explain it.

Table 3:

Same that in table 2

Table 4:

Same that in table 2

Discussion:

Line 323-324

“chronological age, and the interaction between chronological age and relative age” interaction is not included anywhere.

Line 328:

Citation 34, this paper is about female sport, is not enough to explain what are you relating.

Lines 337-338

“i.e., the interaction did not meet the criteria for statistical significance 337 [p > .05] in any of the participation/non-participation contexts” Where is it in results chapter?

Line 350:

Citation 26 has important limitations; it is just about skiing, figure skating and gymnastic. i.e. evidence in citation 28 is stronger.

Line 353:

“technical skill have not produced equivalent findings” Athletics relay high levels of technical skills.

Lines 360-362:

“While “individual sport” in this study included several activities associated with physical demands, the lack of evidence for biased birth distributions suggests this type of sport context may be more likely to provide equitable opportunities for success” This is a clear limitation of the study, you need to remark this question.

Lines 366-368:

“i.e., a stronger risk of RAEs would be expected at elite levels of performance and may be somewhat diminished in a heterogenous sample” there are evidences in Basquetball in the international review for the sociology of sport (doi: 10.1177/1012690212462832).

Conclusion

Line 439-440:

“while individual sport contexts appeared to be unaffected” Discussion is not enough about this question

Line 444:

“Avenues for future research” this is not a conclusion

Author Response

Thank you for your time and recommendations for our paper.  Please refer to the attached document for a detailed outline of changes.

Reviewer 2 Report

The paper is interesting and in general well written

I have some minor comments mainly concerning relatively old cited papers and adding some more fresh in terms of citing relevant works

TITLE – need to be shortened

ABSTRACT – need to be shortened

Introduction – need to be shortened

1.     Intro need more fresh articles to be cited

It would be worthy to mention in the first paragraph that PA is in general related with prevention against incorrect body posturÄ™ among young people however some sports may enhance incorrect body posture

You can citate papers below

Material and methods

This part is correct and well written

Discussion

I suggest you to include the following work in the discussion

http://tss.awf.poznan.pl/files/2020/Vol%2027%20no%202/3_Maciaszek_TSS_2020_272_63-69.pdf

Although the suggestion below is for students, perhaps it is worth referring to it in the discussion, it is fresh, much fresher than what is cited in the paper

http://tss.awf.poznan.pl/files/2020/Vol%2027%20no%201/5_Wasiluk_TSS_2020_271_29-34.pdf

Author Response

(The authors gave the same response as above.)

Reviewer 3 Report

Thank you for the opportunity to review your work.
The topic is very essential in the science of physical culture.
I rate the quality of your work highly.
The article is technically correct and internally compatible.
I think the external reception will be very positive.

The topic is very essential in the science of physical culture. The study examines the patterns of longitudinal participation in organized team and individual sport in a sample of adolescents and brings important innovations in this area.   I rate the quality of your work highly. The article is technically correct and internally compatibleThe strong point is a large research sample from Canada, verified three times a year. The multi-level logistic models of analyzes are correctly selected and implemented. The structure of the work, the division of content and conclusions are thoroughly carried out. I have not noticed the weaknesses of this work.   My positive assessment is justified by the internal consistency of the reviewed work. The language and style of the authors is at a high level. Moreover, the authors showed research maturity. References are selected correctly and impartially. Therefore, I think that the external reception of this work will also be positive. There is little research on this subject in Europe. Therefore, this research may constitute an important reference point for future directions of research in the sciences of physical culture.

Author Response

(The authors gave the same response as above.)

Round 2

Reviewer 1 Report

Thank you very much for your comments and your patience. I agree most of your changes, and even the non accepted suggestions. But still there are a point I can not accept. I suggested you to change the word quartile, in statistics, a quertile is a type of quantile which divides the number of data points in for parts or quarters. There are just 3 quartiles and 4 quarters. Even you feel the use of ‘quartile’ is more appropriate, it is not. You need to change it by quarter or trimester, as you wish, but not quartile.

I hope you understand this request, I consider it is very important.

Kindest Regards and congratulations about your paper, it is really interesting and you answered all my other requestings

Author Response

We have replaced the term "quartile" with "quarter" throughout the manuscript.  Thank you for your review of our paper.